# A New Kid on the Block? Carbonic Anhydrases as Possible New Targets in Alzheimer’s Disease

**DOI:** 10.3390/ijms20194724

**Published:** 2019-09-24

**Authors:** Gustavo Provensi, Fabrizio Carta, Alessio Nocentini, Claudiu T. Supuran, Fiorella Casamenti, M. Beatrice Passani, Silvia Fossati

**Affiliations:** 1Department of Neuroscience, Psychology, Drug Research and Child Health (NEUROFARBA), Section of Pharmacology of Toxicology, University of Florence, 50139 Florence, Italy; fiorella.casamenti@unifi.it; 2Department of Neuroscience, Psychology, Drug Research and Child Health (NEUROFARBA), Section of Pharmaceutical and Nutraceutical Sciences, University of Florence, 50019 Florence, Italy; fabrizio.carta@unifi.it (F.C.); alessio.nocentini@unifi.it (A.N.); claudiu.supuran@unifi.it (C.T.S.); 3Department of Health Sciences (DSS), Section of Clinical Pharmacology and Oncology, University of Florence, 50139 Florence, Italy; beatrice.passani@unifi.it; 4Alzheimer′s Center at Temple (ACT), Department of Pharmacology, Lewis Katz School of Medicine, Temple University, Philadelphia, PA 19140, USA; silvia.fossati@temple.edu

**Keywords:** aging, neurodegenerative diseases, cognition, oxidative stress, mitochondria

## Abstract

The increase in the incidence of neurodegenerative diseases, in particular Alzheimer’s Disease (AD), is a consequence of the world′s population aging but unfortunately, existing treatments are only effective at delaying some of the symptoms and for a limited time. Despite huge efforts by both academic researchers and pharmaceutical companies, no disease-modifying drugs have been brought to the market in the last decades. Recently, several studies shed light on Carbonic Anhydrases (CAs, EC 4.2.1.1) as possible new targets for AD treatment. In the present review we summarized preclinical and clinical findings regarding the role of CAs and their inhibitors/activators on cognition, aging and neurodegeneration and we discuss future challenges and opportunities in the field.

## 1. Introduction 

Alzheimer’s disease (AD) is the most frequent type of dementia. Dementia, an umbrella term to define a collection of symptoms including communication, thinking and memory impairments, has been declared by the World Health Organization (WHO) through the Mental Health Gap Action Programme, as a priority condition. According to the WHO, the costs globally committed for treating and caring people with dementia are more than 604 billion U.S. dollars per year. The projections emerging from current data on the incidence and prevalence of dementia indicate that the number of people affected will continue to increase, thus the associated budgets are likely to increase. Dementia associated costs in Europe are rising sharply (43% from 2008) and the estimates indicate that they will reach 250 billion euros in 2030 [1]. The global societal cost for dementia is expected to reach 2 trillion U.S. dollars in 2030 [2]. The WHO′s report has been recently confirmed by the Alzheimer’s Disease International research which has estimated that every 20 years the number of patients will be doubled, reaching approximately 65.7 million in 2030 and up to 115.4 million in 2050 [3]. 

Main clinical symptoms of AD include gradually worsening ability to remember new information and global cognitive deficits that can lead to dementia with the disease progression and non-cognitive symptoms, especially loss of motor functions, gait disturbances, disturbed balance. The main pathological features of AD—amyloid-β (Aβ) plaques, neurofibrillary tangles (NFTs), astrogliosis and neuronal loss—were described by Alois Alzheimer in 1906 [4]. Microgliosis, inflammation, oxidative stress, major synaptic alteration and cerebral amyloid angiopathy are other pathological hallmarks of AD [5,6,7,8]. The sequential cleavage of amyloid protein precursor (APP) by β- and γ-secretases originates the Aβ peptide. Even though the aetiology of AD is not completely understood, the “amyloid hypothesis” indicates a central role for Aβ not only in plaques formation but also in the cascade leading to the other pathological hallmarks of the disease including tangle formation and neuronal cell death [9]. Based on this hypothesis numerous animal models, diagnostics and therapeutics for AD were generated. However, the amyloid hypothesis has been recently questioned by some authors. Nonetheless, prevention is still considered a valid strategy to avoid or delay the onset of neurodegenerative diseases characterized by amyloid deposits. In this regard, hindering amyloid aggregation and subsequent plaque deposition can be achieved with both pharmacological and lifestyle strategies. To date, there is no effective treatment for AD and current therapeutic strategies only alleviate its symptoms and neither modify the underlying disease nor delay its progression. The goal of future therapies should be to improve or at least to maintain the patients’ baseline performances through the treatment with disease-modifying drugs. Accordingly, researchers are looking for new multi-target drugs and combination therapies to treat AD, including anti-inflammatory, anti-amyloid and anti-oxidant approaches. 

Oxidative stress has been considered one of the mechanisms underlying AD pathology and an unbalance between oxidants and antioxidants may result in increased reactive oxygen and nitrogen species leading to oxidative damage to several biological molecules. Oxidative-induced protein modifications may alter their functions, including their catalytic activity [10]. For instance, a decrease of carbonic anhydrases (CAs) activity and a series of proteins excessively nitrated and/or carbonylated, including the isoform CA II, have been described in the AD hippocampus [11,12,13,14] and in brain samples obtained from mild cognitive impairment patients (MCI), suggesting that the increase in oxidative modifications dropped the enzyme catalytic activity during the preclinical AD stages [15]. Moreover, CA II has been identified among numerous abundant plaque proteins, suggesting that it may play a central role in plaque development or co-occur with plaque formation [16]. The high CA II levels found in central [13,17] and in peripheral systems [18] also suggest the possibility that CA II expression may represent a biomarker for AD, as we will discuss bellow. Furthermore, promising preclinical evidence using CA inhibitors (CAIs) in models of amyloidosis has also been recently reported [19]. These findings, as well as future challenges and perspectives in the field, will be discussed in this review.

## 2. Carbonic Anhydrases in the Central Nervous System 

The reversible hydration of CO_2_ is a basic reaction assuming a paramount importance in carbon-based life, operating in water-based media. All forms of life on Earth share the same biochemistry, which is based on the wealth of chemical transformations of “carbon”. The conversion of the latter into its biologically fruitful form, as well as the opposite reaction, are represented in Equation (1).
CO_2_ + H_2_O ⇋ H^+^ + HCO_3_^−^(1)

This reaction has been kinetically investigated and proven to occur significantly below the threshold required for the biochemical transformations which maintain the evolution of life. In this context are the CA enzymes, which have the role to speed up such reaction up in order to properly match the biological needs. The abundance of genetic families and CA isoforms expressed within, undoubtedly reflects their outstanding biological value (i.e., 15 isoforms have been reported in humans so far) [20]. As a consequence, CAs assume a privileged role in Medicinal Chemistry being the ideal targets for the management of various diseases including those affecting the CNS [20,21,22]. Almost all 15 CA isoforms have been identified in the CNS or the choroid plexus [23]. The cytosolic and ubiquitous CA I is expressed in the motor neurons in human spinal cord [24]. The physiologically dominant isoform CA II is located both in the choroid plexus and in oligodendrocytes, myelinated tracts, astrocytes and myelin sheaths in the vertebrates brain [25]. The choroid plexus also contains CA III which presumably possesses an unknown physiological function because of its weak CO_2_ hydration catalytic activity [26]. CA IV, that is anchored to the cell membrane by a glycosylphosphatidylinositol tail, is located on the luminal surface of cerebral capillaries, associated with the blood-brain barrier and expressed in layers III and VI in the cortex, thalamus and hippocampus [25,27]. Immunocytochemical experiments demonstrated that astrocytes and neurons express the mitochondrial CA VA suggesting that this isozyme has a cell-specific, physiological role in the nervous system [28]. There is instead a lack of studies on CA VB in the brain, although it was recently detected in rat middle cerebral arteries [29]. CA VII is endowed with a good CO_2_ hydrase activity and CA VIII is instead acatalytic—the two isoforms show comparable expression in the cortex, hippocampus and thalamus [30,31]. CA VII might be considered a brain-associated CA as it is absent in most other tissues [30]. CA VII is considered a key molecule in age-dependent neuronal pH regulation [30]. The acatalytic CA X and XI are located in the myelin sheath and in the neural cell body and astrocytes of few brain regions, respectively [31]. CA IX and XII are generally overexpressed in tumours and high levels were in fact identified in many neurologic cancers such as ependymoma, glioma, meningioma, hemangioblastoma and choroid plexus tumours [25,32]. Nonetheless, high levels of CA XII were also observed in the choroid plexus [33]. High levels of CA XIV were detected in neuronal bodies and axons in the anterolateral part of pons and medulla oblongata as well as in corpus callosum, hippocampus, cerebellar white matter and peduncles, pyramidal tract and choroid plexus [34]. CA XIII has not been identified in the brain to date. 

Epithelial cells of the choroid plexus produce cerebrospinal fluid (CSF) by a process implicating the transport of Na^+^, Cl^−^ and HCO_3_^−^ from blood to the brain ventricles [35] which induces an osmotic gradient responsible of water secretion into the CSF. HCO_3_^−^ transporters, such as anion exchangers (AEs), Cl^−^/HCO_3_^−^ exchangers, Na^+^/HCO_3_^−^ cotransporters, have a role in this process, as well as CA isoforms that choroid plexus express in rather high amounts such as CA II, CA III and CA XII [35]. 

The peripheral nervous system also retains CA activity, whose distribution was assessed histochemically in rat cross-sections of peripheral nerves [36]. Intense CA activity was mainly detected in large diameter (8–12 microns) muscle afferents. Another study detected high CA activity in large nerve fibres in cross-sections of the infraorbital branch of the fifth cranial nerve [37]. To date, there are no evidences of the type of CAs present in the peripheral nervous system. 

## 3. The Role of Carbonic Anhydrases in Cognition

The first evidence supporting a role for CAs on memory processing was given by the group of Dr. Miao-Kun Sun and Dr. Daniel L Alkon at the Blanchette Rockefeller Neurosciences Institute, Rockville, Maryland back in 2001. They demonstrated that a single systemic injection of acetazolamide (AAZ), a CA inhibitor (CAI) that penetrates the blood brain barrier (BBB), administered one hour before the first trial was sufficient to produce a significant impairment in spatial memory as measured in the Morris Water Maze (MWM) task [38]. In the same year, these researchers also reported that spatial learning of rats infused into the lateral ventricles with D-phenylalanine (D-PHE), a CA activator (CAA), was faster than in animals receiving saline infusions [39]. Co-infusions of AAZ not only prevented D-PHE-induced procognitive effects but it also caused memory impairments. This observation excluded the involvement of other D-PHE mechanisms, such as increased catecholamine biosynthesis and/or transmission [40]. Noteworthy, swim speeds did not differ between groups, indicating that CA activation or inhibition did not affect- significantly sensory or locomotor activities [38,39]. In agreement with these observations, CA IX-knock out mice showed impairments in the MWM test when compared to wild type animals [41].

The negative impact of CA inhibition on emotional memory is also reported. Systemic acute administrations of AAZ impaired animals’ performance in a dose dependent manner in two fear-motivated paradigms. In the passive avoidance test, mice receiving AAZ, either before (1 h) or after (30 min or 2 h) training, showed shorter step-through latency when compared to the animals treated with vehicle. This behaviour was evident when the retention test was performed 24 h later. In the active shuttle avoidance test, whereas control rats showed a clear fear memory learning characterized by increased avoidance response throughout the training days, the overall avoidance responses were significantly inhibited in rats treated with AAZ 1h before avoidance task. Treatment with AAZ did not alter startle response to either sound stimulation or electric shock, thus ruling out the possibility that such effects may be related to alterations in rodents’ sensitivity [42]. Taken together these findings strongly suggest that acute CA inhibition impairs fear memory acquisition and consolidation.

The effects of the modulation of CA activity was also evaluated using an instrumental conditioning test [43]. During the first phase of this paradigm, male mice were placed in the apparatus and a tone was delivered in a variable-time schedule for 6s. If during the tone presentation the animal executed a nose-poke response, it received a reinforcer (a hypercaloric flavoured solution) and the tone was turned off. This session lasted 1h or until the animals obtained 10 reinforcers. Drugs were given i.p. immediately after the end of this session and 1 h later the animals were submitted to a second session, with the same procedure. A reduction in the latency to earn the rewards or increase in the reinforced responses or decrease in the non-reinforced responses are interpreted as learning. AAZ systemic treatment dose-dependently augmented latencies and reduced reinforced responses similarly to scopolamine, an amnestic agent. The authors also studied, in the same task, the effects of an ethylene bis-imidazole derivative acting as a relatively selective CAA (CA V and VII), expecting a learning improvement. However, in doses up to 30 mg/kg given either 30 min before or immediately after the first session did not alter significantly the behaviour or the animal during the test [43]. These data demonstrate that while acute inhibition of CA evidently impairs instrumental conditioning learning, further studies are needed to elucidate the impact of CAAs on this complex learning task.

Recently, some of us demonstrated the participation of brain CAs in recognition memory using the novel object recognition test in mice. We found that systemic administration of D-PHE, significantly augmented CA activity measured in brain homogenates and improved animals’ performances in the memory task. On the contrary, AAZ caused memory impairments which parallels with significant reduction of CAs activity in the brain. Systemic administration of a positively-charged, membrane-impermeant pyridinium sulphonamide (compound 18 [44]) which is unable to cross the BBB, did not affect memory formation. Consistently, co-administration of AAZ with D-PHE fully prevented the CAA-induced memory improvement, whereas co-administration of compound 18 did not affect D-PHE-induced procognitive effect. This series of results clearly indicates that the modulation of central CAs is involved in learning and memory processing [45].

Evidence emerging from clinical studies further supports a role of CAs on cognition. A randomized, placebo controlled, double-blind trial evaluated the effects of AAZ treatment on cognitive neuropsychological measures during acute high-altitude exposure in healthy volunteers. The study revealed that, compared with the placebo-treated group, subjects receiving AAZ showed impairments on cognitive processing, short-term and working memory, attention and concentration [46]. Though, in recent clinical trials of other conditions such as idiopathic intracranial hypertension [47] or sleep apnoea [48,49], doses of up to 2g/day of AAZ were given and typically well tolerated by patients, with no reports of negative effects on cognition. Further evidence correlating the inhibition of CAs and memory deficits in humans has emerged from studies on topiramate, an anticonvulsant drug that acts also as a CAI [50,51]. Epileptic patients receiving topiramate treatment presented significant cognitive impairment that regressed when the drug was discontinued or the dose reduced [52,53]. However, topiramate is currently used for migraine, seizures, alcohol use disorders, addiction, post-traumatic stress disorder and other psychiatric and neurological disorders with positive effects. An administration strategy beginning with 25 mg once daily for 1 week and gradually titrating to twice daily doses of up to a maximum of 200 mg is now used in these pathologies and titration of the drug has been shown to limit negative cognitive effects [54,55,56]. It is important to note that besides inhibiting CAs, topiramate has multiple pharmacological effects including potentiation of GABA transmission, antagonism of glutamate receptors, alteration of voltage-gated calcium and sodium channels [57] that must be considered in the interpretation of these findings. 

The mechanisms underpinning CAs actions on cognition remain largely unidentified. Modulation of CAs activity alters the buffering capacity, thus influencing intracellular and extracellular pH value, therefore affecting protein NMDA and GABA receptors function [58]. Early studies demonstrated that the associated activation of multi-synaptic inputs on pyramidal neurons in the hippocampal CA1 region transiently transform GABAergic IPSPs to EPSPs; this transformation depends on CA activation as it busts HCO_3_^−^ intracellular concentrations favouring its efflux through the GABA_A_ receptor channel [39,59]. Therefore, GABA-mediated responses on CA1 pyramidal neurons become excitatory amplifying synaptic weights relevant to a particular memory processing (reviewed in Reference [60]). Therefore, CAs acts as a gate potentiating signal transfer through the neural network. 

More recently, it has been demonstrated that treatment with D-PHE augmented significantly ERK 1/2 phosphorylation in hippocampal and cortical homogenates. Such effect was prevented by the co-administration of AAZ, whereas the co-administration of compound 18, a brain-impermeant CAI, did not affect D-PHE-induced effect [45]. These findings were in agreement with previous reports showing that increased ERK 1/2 phosphorylation in the amygdala due to fear conditioning training was also inhibited by AAZ treatment [42]. The genomic response activated by ERK pathway [61,62] is an essential step for consolidation and persistence of several types of long-term memories [63]. Therefore, CAs-induced increased ERK phosphorylation is necessary for memory consolidation.

## 4. Effects of Carbonic Anhydrases Inhibitors in Preclinical Models of Neurodegenerative Diseases 

Intriguingly, despite the acute amnesic effects of CAIs, cellular and animal models of amyloidosis have shown that these compounds can be protective against mitochondrial dysfunction, oxidative stress and prevent memory loss induced by amyloid aggregates. The present paragraph will discuss these findings. The mechanisms are summarized in Figure 1. Methazolamide (MTZ) and AAZ are Food and Drug Administration (FDA, USA)-approved CAIs used chronically in humans for treatment of glaucoma, prevention of high-altitude sickness, high altitude cerebral oedema and other indications, including seizures. Interestingly, MTZ was identified, among more than 1000 compounds from the National Institute of Neurological Disorders and Stroke (NINDS) drug library, as one of the few drugs able to inhibit Cytochrome C (CytC) release from purified mitochondria in vitro [64]. It was also demonstrated to be neuroprotective in cell and animal models of Huntington’s disease (HD) and stroke [64,65]. Studies conducted by some of us tested for the first time MTZ, as well as its analogue AAZ, which is more widely used in clinical settings, in in vitro and in vivo models of amyloidosis [66,67,68,69]. We showed that both compounds are effective at preventing mitochondrial dysfunction, caspase activation and cell death. In our hands, AAZ was 10 times more effective than MTZ at inhibiting the same toxic mitochondrial mechanisms induced by Aβ in neuronal and vascular cells in vitro [67].

The mechanisms of action responsible for the protective effects of CAIs against neurovascular pathology can be manifold and may be related to their ability to interfere with the production of ions, the activity of different pumps and, in the CNS, to minimize abnormal and excessive discharge. CAIs are also useful, among other things, to increase cerebral blood flow [70], decrease extracellular pH in the brain promoting vasodilation [71] and increase neuronal excitability [72]. The main mechanism of action pointed out by our studies was the ability of the CAIs to prevent mitochondrial dysfunction mechanisms induced by Aβ. Among the known CAs, CA VA and CA VB are mitochondrial [20] and CA II is increased in the mitochondria during aging and neurodegeneration [17]. We have shown that MTZ and AAZ are effective against Aβ-mediated mitochondrial toxicity and cell death in vitro and in vivo [66,68]. We recently showed that the key mechanisms for these effects are related to the ability of CAIs to reduce overproduction of the mitochondrial reactive oxygen species (ROS) hydrogen peroxide (H_2_O_2_) and prevent the associated loss of mitochondrial membrane potential (ΔΨ) and mitochondrial CytC release, contributing to the reduction of caspase activation and neuronal and endothelial cell death in our models [67]. Although none of the presently available CA inhibitors has a specific isozyme selectivity [20], both MTZ and AAZ have high affinity for the mitochondrial CA enzymes, suggesting that the mitochondrial effects may be driven by inhibition of CA VA and VB. Additional work to elucidate these questions is in progress.

In addition to the prevention of mitochondrial dysfunction and cell death, in our studies MTZ and AAZ appear to be also effective to reduce memory impairment and amyloid pathology in a transgenic mouse model of amyloidosis (TgSwDI) [19]. Blocking early mitochondrial damage and H_2_O_2_ production using CAIs may represent an effective approach for prevention and therapy of AD. CAIs such as AAZ are well tolerated in humans up to a 1000 mg daily dose and already FDA-approved for the prevention of acute mountain sickness and consequent high-altitude cerebral oedema, confirming that these drugs are effective in the brain and safe for systemic administration [73]. 

As an additional positive effect, CAIs are known to promote vasoreactivity and increase cerebral blood flow. Cerebral blood flow is increased in patients treated with AAZ and tissue oxygenation is improved [74,75]. Therefore, these compounds, in addition to the prevention of mitochondrial dysfunction, which is the main mechanism of action found in our studies, may also improve vascular reactivity and activate cerebral blood flow. This additional effect has the potential to result in an increase in brain clearance in vivo, which can reduce the accumulation of toxic aggregates of amyloid and tau in AD animal models and patients. Accordingly, a recent clinical trial employing AAZ for obstructive sleep apnoea (OSA) with comorbid hypertension reported positive effects, with reduction of blood pressure and apnoea-hypopnea index. The reduction of venous bicarbonate concentration following AAZ was correlated with the change of apnoea-hypopnea index. AAZ also reduced vascular stiffness and sleep apnoea [75]. This is of particular interest for the field of dementia and neurodegenerative diseases, as both hypertension and OSA are important comorbidities in AD and mixed dementia patients. The FDA-approved systemic use of CAIs for glaucoma therapy and prevention of high-altitude sickness, among the other applications, also suggests safety for chronic treatments. Moreover, the pharmacokinetics and pharmacodynamics in animals and humans have been already tested [20,76,77] and it is known that CAIs can easily cross the blood brain barrier. In recent clinical trials, doses of up to 2g/day of AAZ are given and typically well tolerated by patients. The main side effects reported are mild paraesthesia and dyspepsia, with no report of negative cognitive effects. 

With regard to brain vessel wall effects, a recent study by Rasmussen and colleagues on isolated rat middle cerebral arteries showed that switching from CO_2_/HCO_3_^−^ free to CO_2_/HCO_3_^−^ containing bath solution at a fixed pH of 7.4 caused a biphasic pH response with initial acidification followed by a gradual recovery of pH and vasoconstriction [29]. In the presence of AAZ the rate of intracellular acidification and vasoconstriction were reduced, whereas applying the cell-impermeable CAI AMB (4-(aminomethyl)benzenesulfonamide) did not affect the rate of intracellular acidification. However, CA inhibition did not affect overall acid extrusion from vascular smooth muscle cells. This study also shows that CAs such as CA VB are expressed and have important functions in the cerebral vessel wall and that intracellular CA activity accentuates pH transients and vasoconstriction in response to acute elevations of pCO_2_, whereas AAZ attenuates the transient CO_2_-mediated vasoconstriction. Since Aβ has been also shown to have vasoconstrictive effects [8], CAIs may help limiting vasoconstriction in the AD brain, allowing improved perfusion and clearance [78]. 

## 5. Altered Carbonic Anhydrases Expression and Activity in Neurodegenerative Diseases—Clinical Evidences

In the last two decades, studies demonstrating alterations in CAs expression and catalytic activity in post-mortem brain tissue obtained from patients with AD and MCI started to accumulate. The main findings of these studies are summarized in the next paragraphs and in the Table 1.

Recently, considerable efforts in AD research focused on modifications in the BBB that may affect Aβ peptide clearance. In this context, increasing attention has been given to the participation of blood-CSF barrier and CSF itself in the pathophysiology of AD. For instance, during early AD stages CSF hydrostatic pressure rises and then in moderate-to-severe stages it decreases when compared to normal subjects, due to a reduction of CSF production and turnover [79,80,81]. Such a decrease may negatively impact the removal of toxic metabolites from the brain, including Aβ peptide. CSF is mainly produced (nearly 75%) in the four choroid plexi (CP) present in the brain ventricles [82]. Aiming to elucidate the genetic factors related to the reduced CSF secretion and turnover in AD, the group of Prof. Gerald D. Silbenberg at Stanford University investigated, in post-mortem tissue of 6 control and 7 late-stage AD patients, the mRNA expression of a panel of genes hypothetically related with CP integrity and CSF production, including CAs. Interestingly, a particular expression pattern of the several CA isoforms was found—CA II, CA III and CA IV were downregulated, whereas CA XIII was up-regulated. The other CAs gene transcripts were not significantly altered, however a tendency towards downregulation was observed for CAV III and CA XIV and a tendency for upregulation was found to CA VII, CA VB and CA XI [83]. Bicarbonate (HCO_3_^−^) ions produced during the carbon dioxide hydration catalysed by CAs sustain CSF production, since HCO_3_^−^ gradients are used by several transporters to move ions across the CP epithelium [84]. Downregulated CAs could directly impact on HCO_3_^−^ equilibrium, diminishing solute transport and therefore reducing CSF production. In keeping with this hypothesis, it was observed that inhibition of CAs with AAZ reduces CSF secretion by half in rats [85]. However, clearance of brain products is driven by both CSF movements and vascular pulsatility [86,87]. AAZ is clinically used to increase cerebral blood flow and vasoreactivity [88,89]. This effect, promoting perivascular clearance through increased vascular flow and pulsatility, which drives movement of perivascular fluids, may be beneficial for brain clearance of waste products such as Aβ [78,86]. 

It is important to note though, that while changes in gene expression are good indicators, they do not always have a functional impact, since mRNA levels and protein expression are not necessarily strictly related. In this regard, the group of Prof. D. Allan Butterfield at the University of Kentucky, performed a series of proteomic studies with frozen post-mortem tissue obtained from AD patients and age-matched controls (six in each group). Samples were adequately prepared, submitted to a two-dimensional electrophoresis separation and finally the protein spots were recognized using mass spectrometry. The expression of several proteins was altered in the hippocampus of AD samples, including CA II, which was found to be increased [13]. In neurodegenerative disorders, particularly in AD, the antioxidants and oxidants unbalance can result in increased reactive nitrogen and oxygen species which may lead to oxidative damage to several biological molecules. Oxidative-induced protein modifications may alter protein functions, including their catalytic activity [10]. Therefore, the authors explored, using the same approach and in the same samples, the levels of oxidized proteins. Interestingly they found again a series of proteins excessively nitrated and/or carbonylated in AD hippocampus when compared with controls. Among these modified proteins, CA II was again identified [11,12]. A small decrease in CA activity (around 20%) was also found in AD samples [11]. A similar reduction in CA catalytic activity was reported earlier in the temporal lobe of AD patients [14]. Taken together these observations suggest that although CA II protein levels are augmented in AD hippocampus, the oxidative modifications (carbonylation/nitration) may have negatively impacted on its function and therefore decreased CA catalytic activity when compared with normal subjects. Interestingly, these alterations were found in the hippocampus, a brain region showing high levels of Aβ peptide deposition but they were not found in the cerebellum, where Aβ levels were negligible [90], suggesting that modifications of CA expression and activity may be correlated with Aβ deposition. In keeping with this hypothesis, Drummond and colleagues, performed a localized proteomics study aiming to investigate the protein differences between amyloid plaques microdissected from the hippocampus and adjacent entorhinal cortex of sporadic and rapidly progressive AD patients (22 patients for each group). Across the samples, 1934 different proteins were identified and despite minor inter-subject variations, significant differences of the proteomes were found between groups. Of note, 279 of these proteins were present in plaques from all samples examined. CA II was identified among these abundant proteins detected in all patients, suggesting that these enzymes an important role in plaque maturation or co-occur with plaque formation. However, further studies are needed to support this hypothesis [16]. 

The oxidative stress hypothesis has gained great attention as one of the mechanisms underlying AD pathology. However, an important unsolved question is whether oxidative stress is the cause of neuronal dysfunction, thus contributing to the pathogenesis and progression of AD or it is a result of neurodegeneration. In this regard, again using two-dimensional electrophoresis associated with mass spectroscopy, oxidatively modified proteins were studied in samples obtained from patients with MCI (6 samples) and in age-matched control subjects (4 cases). Four proteins were found to be significantly more oxidized in the inferior parietal lobule of MCI samples and CA II was among these [15]. CA activity was also reduced in these samples (about 50%), suggesting that the oxidative damage dropped the catalytic activity early on during disease progression [15]. 

The levels of plasma CA II were also analysed in a large study including 91 AD patients, 83 MCI cases and 113 cognitively age-matched normal controls. CA II concentration in plasma samples was determined using Western blot and ELISA techniques. The CA II expression was found to be higher in the plasma of AD patients than in the control group. The CA II levels determined in the MCI group were slightly higher than those of control subjects but lower than AD patients but these differences did not achieve the threshold for statistical significance. Interestingly, a positive correlation was also observed between aging and CA II levels, which was particularly prominent in the AD group. When the samples were stratified by gender, CA II levels were found to be slightly higher in males than in females in all groups [18]. Increased CA II levels found centrally and also in the periphery suggest the possibility that CA II expression may represent a biomarker for AD. However, its adequacy as a biological marker depends on further studies clarifying if this enzyme has a role on AD pathogenesis or is a consequence of the disease progression [18]. 

Not all data, though, support the hypothesis of CA alterations in AD. For instance, the group of Prof. Gert Lubed at the University of Vienna, investigated CA II expression in brain tissue obtained from patients diagnosed with AD (11 cases), Down Syndrome (DS, 9 cases) and age-matched controls (14 subjects) [91]. DS patients were included in the study because the amyloid-protein precursor (APP) is encoded by a gene located on chromosome 21, consequentially trisomy 21 results in APP overexpression and higher Aβ plaque deposition. As a result, practically all DS patients over 40 years show abundant neuritic plaques and neurofibrillary tangles [92]. However, this study did not disclose any significant alteration in CA II expression in the caudate nucleus, thalamus, cerebellum or cortical regions (frontal, temporal, occipital and parietal) of DS or AD brains in relation to the samples obtained from control subjects [91]. More recently though, increased CA II levels were found in the frontal and temporal cortices of infants with DS and also in the brain of Ts65DN mice, a murine model of DS [93]. These findings suggest that CA II levels may be increased just in early periods of life in the brain of DS patients [93]. 

## 6. Challenges and Perspectives on Carbonic Anhydrases Research and Therapeutic Exploitation

In the past decades, the knowledge regarding the biology that underlies AD has advanced tremendously thanks to the enormous efforts undertaken by the scientific community around the world. However, AD (as well as other neurodegenerative diseases) is still a clinical challenge. Notwithstanding the economic, healthcare and social burden of AD, at present no disease-modifying treatments are available. Current clinically used drugs are useful in slowing the progression of disease symptoms but the benefits of the therapy hardly extend beyond an average of 6 months and unfortunately the therapy is ineffective in many patients [94].

Despite the unmet medical need for effective treatments for AD, almost no drug has been brought to the market in the last decade [95]. Indeed, clinical trials of drugs targeting amyloid and tau did not meet the expectations. This emphasizes why further increasing our knowledge of the underlying mechanisms of AD pathology in order to identify new targets for the development of innovative pharmacological tools is critical. The development of original and effective therapies will have a huge impact on public health. For example, it is estimated that a treatment producing a modest one-year delay in the AD onset by 2020 would drop the number of new cases by 9.2 million in 2050 [96]. As stated in this review, brain CA may represent an innovative target.

CAs are a family of ubiquitously expressed metalloenzymes. In humans, 15 different isoforms were identified and several of them are present in the CNS, with different sub-cellular localization, catalytic activities and expression pattern among the brain areas [20]. Despite the high CA expression in the brain, their functions are still not fully understood [97]. CAs catalyse the carbon dioxide hydration producing protons and bicarbonate ions, thus playing a pivotal role in physiological processes such as brain pH control, CSF production and neuronal excitability [98]. These enzymes are also involved in pathophysiological conditions and indeed CAIs are currently used as anticonvulsant agents [99] and in the treatment of idiopathic intracranial hypertension [100]. Recent studies are revealing the efficacy of these compounds for other CNS-related disorders such as obesity [101], brain ischemia [102], neuropathic pain [103], migraine [104], sleep apnoea [105] and so forth (reviewed in Reference [97]).

Recently, a series of observations are shedding light on the potential involvement of CAs on neurodegenerative disorders, particularly in AD—(i) increased CA II was found in the mitochondria during aging and neurodegeneration as revealed by proteomic studies [17]; (ii) CAs inhibition prevented Aβ-induced mitochondrial dysfunction related to neuronal and microvascular toxicity in vitro and in the mouse brain [19,66,67]; (iii) CA II is found in amyloid plaques in the AD human brain [16]; (iv) CA II expression is increased in hippocampus and in plasma of AD patients [13,18]; (v) alterations of CA catalytic activity were reported in the temporal lobe and hippocampus of AD post-mortem samples [36,37]; (vi) CA inhibition is used as a strategy to increase cerebral blood flow and vasoreactivity in multiple clinical trials [88,89]. Even if these findings may seem apparently easily interconnected, is clear that at this point many pieces are still missing to complete the puzzle. For instance, encouraging data show that AAZ prevents Aβ-toxicity in vitro [66,67] and a single systemic injection of AAZ prevented neurodegeneration induced by Aβ-peptide hippocampal infusion [66]. On the other hand, reduced CA activity was found in post-mortem AD brain tissue [11,14,15] and AAZ acute administration was found to induce amnesic effect in rodents [42,45] and humans [46]. It is important to note though that recent clinical trials using this drug even in very high doses did not reveal any cognitive impairments [29,30,31]. Therefore, due to these contradictions between preclinical and clinical findings and acute versus chronic administrations, if the best therapeutic strategy would be to use activators or inhibitors and during what phase of disease progression is still not clear.

Preclinical studies are essential to clarify these inconsistences and fill the gap between the available in vitro and post-mortem human studies. In this regard, current studies by some of us (Dr. Fossati’s group) are elucidating the protective effects of long-term treatment with CA inhibitors in transgenic mouse models of cerebral amyloidosis [19]. However, several other issues must be addressed—for instance, a comprehensive description of the expression and catalytic activity of the different isoforms during AD progression will help to identify specific molecular targets to concentrate medicinal chemical efforts for the development of specific CAA/CAI. Another important aspect is to evaluate the putative preventive/therapeutic effects of different CA modulators in animal models of amyloidosis. Since some of these compounds, either activators (D-PHE) or inhibitors (i.e., AAZ, topiramate) are already used in humans, the potential positive effects observed in animal models could pave the way for fast-track off-label clinical trials. 

## Figures and Tables

**Figure 1 ijms-20-04724-f001:**
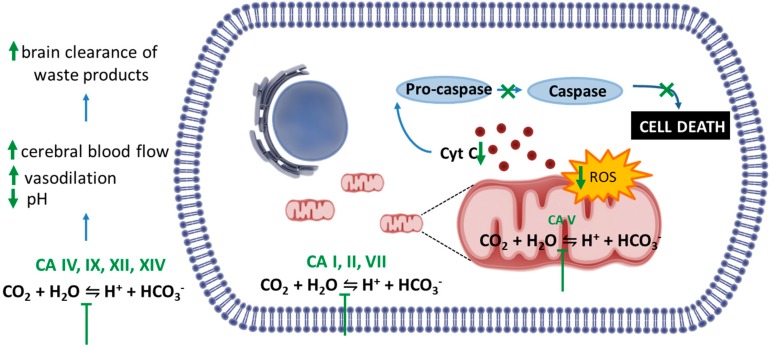
Potential mechanisms underlying the effect of Carbonic Anhydrases inhibitors (CAIs) in preclinical models of amyloidosis. Studies in vitro and in vivo highlighted the efficacy of CAIs in preventing Aβ-induced mitochondrial toxicity and cell death. These compounds are able to reduce overproduction of the mitochondrial reactive oxygen species (ROS) and prevent the associated loss of mitochondrial membrane potential and mitochondrial Cytochrome C (CytC) release, contributing to the reduction of caspase activation and neuronal and endothelial cell death induced by Aβ oligomers. Other possible mechanisms include the decrease extracellular pH in the brain, which in turn induce vasodilation, increase in cerebral blood flow and vasoreactivity. These effects may contribute to the brain clearance of waste products, including Aβ peptide.

**Table 1 ijms-20-04724-t001:** Clinical evidences of Carbonic Anhydrases alterations in Alzheimer’s Disease.

Population	Sample	Brain Area Analysed	Main Findings	Reference
AD patients (sex NI) Age: 80.7 ± 1.7 Control subjects (sex NI) Age: 74.7 ± 2.6	post-mortem brain tissue	temporal lobe	↓ CA activity (around 20%)	[14]
AD patients (7♂, 4♀) Age: 59.55 ± 6.1 DS patients (6♂, 3♀) Age: 55.7 ± 7.48 Control subjects (10♂, 4♀) Age: 56.1 ± 6.1	post-mortem brain tissue	caudate nucleus, cerebellum, thalamus, cortical regions (frontal, temporal, occipital and parietal)	No significant changes in CAII protein levels were found in either DS or CA group compared with controls	[91].
AD patients (4♂, 2♀) Age: 84.5 ± 5.2 Control subjects (4♂, 2♀) Age: 85.8 ± 4.1	post-mortem brain tissue	hippocampus	↑ CA II ↑ CA II oxidized ↑ CA II nitrated ↓ CA activity (around 20%)	[11,12,13]
AD patients (4♂, 2♀) Age: 84.5 ± 5.2 Control subjects (4♂, 2♀) Age: 85.8 ± 4.1	post-mortem brain tissue	cerebellum	No differences were found in CA expression and activity. These findings are described as “data not shown”	[11]
MCI patients (2♂, 4♀) Age: 88 ± 1.5 Controls subjects (2♂, 4♀) Age: 82 ± 2.6	post-mortem brain tissue	Inferior parietal lobe	↑ CA II oxidized ↓ CA activity (around 50%)	[15]
MCI patients (36♂, 47♀) Age: 73.7 ± 5.5 AD Patients (34♂, 57♀) Age: 74.8 ± 7.1 Controls subjects (51♂, 62♀) Age: 70.8 ± 4.6	plasma	-	CA II expression: - AD > MCI > controls - ♂ > ♀ Positive correlation between CAII expression and aging	[18]
sporadic AD patients (10♂,12♀) Age: 70.1 ± 10.3 DD: 9.2 ± 1 month rapidly progressive AD patients (15♂, 7♀) Age: 80.1 ± 9.1 DD: 122 ± 8 month	post-mortem brain tissue	hippocampus and enthorhinal cortex	CA II was detected among the most abundant proteins found in plaques microdissected from all cases examined	[16]
AD patients (7) Control subjects (6) Age and sex NI	post-mortem brain tissue	choroid plexus	↑ CA XIII mRNA ↓ CA II, CA III, CA IV mRNA	[84]

↑ increase, ↓ reduction, ♂ = male ♀ = female, NI = not informed, AD = Alzheimer’s Disease, MCI = mild cognitive impairment, DD = disease duration.

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
