# Peer review of "A New Kid on the Block? Carbonic Anhydrases as Possible New Targets in Alzheimer’s Disease"

_ijms, 2019, doi:10.3390/ijms20194724_

Round 1

Reviewer 1 Report

Recent backlashes in clinical trials in AD have led to a need to humbly take a step back and re-evaluate our current understanding of the disease. The authors of this review paper recognizes this and contributes novel thinking in potential new drug targets. This is important, and the current paper timely adds value to field by reviewing Carbonic Anhydrases as a possible way of treating AD. The paper is well written and, to the best of my knowledge, exhaustive review of this concept. I have only a few minor comments:

On row 360, it is stated that CA II is suggested to have "an important role in plaque maturation". I believe this might be a bit strong, and would suggest a different phrasing of this sentence. Finding CA II in amyloid plaques could possibly suggest a role in maturation, but might also just be co-incidental. Plaques contains many things. On row 401 it is stated that "translational research use valid animal models". Animal models in AD have been to much critique, importantly to the relative ease of finding cures for amyloid pathology in the models vs in humans.  There are several minor spelling and grammatical errors throughout the manuscript that needs fixing.

Reviewer 2 Report

Dear authors,

The paper is well written and i think gives a complete vision of the subject.
I don't think is very innovative but i think there are not comments to add from my side.

Best
